# Do Androids Dream of Electric Fences? Safety-Aware Reinforcement Learning with Latent Shielding

## Abstract

The growing trend of fledgling reinforcement learning systems making their way into real-world applications has been accompanied by growing concerns for their safety and robustness. In recent years, a variety of approaches have been put forward to address the challenges of safety-aware reinforcement learning; however, these methods often either require a handcrafted model of the environment to be provided beforehand, or that the environment is relatively simple and low-dimensional. We present a novel approach to safety-aware deep reinforcement learning in high-dimensional environments called *latent shielding*. Latent shielding leverages internal representations of the environment learnt by model-based agents to "imagine" future trajectories and avoid those deemed unsafe. We experimentally demonstrate that this approach leads to improved adherence to formally-defined safety specifications.

## 1 Introduction

The steady trickle of reinforcement learning (RL) systems making their way out of the lab and into the real world has cast a spotlight on the safety and robustness of RL agents. The motivation behind this should be relatively easy to grasp: when training an agent in real-world settings, it is desirable that some states are never reached as they could, for instance, cause permanent damage to the hardware the agent is controlling. We can thus informally define the notion of *safety-aware RL* in terms of the classical RL setup with the added requirement that the number of *unsafe* states visited be minimised. Under this definition, however, it has been found that many state-of-the-art RL algorithms unnecessarily enter unsafe states despite safe alternatives being available and there being a positive correlation between avoiding such states and reward (Giacobbe et al., 2021).

The field of safety-aware RL encompasses a multitude of approaches ranging from constrained policy optimisation (Chow et al., 2017; Achiam et al., 2017; Yang et al., 2020) to safety critics (Srinivasan et al., 2020; Bharadhwaj et al., 2021; Thananjeyan et al., 2021) to meta-learning (Turchetta et al., 2020). In this work, we focus on a particular family of approaches known as *shielding* (Alshiekh et al., 2018; Anderson et al., 2020; Giacobbe et al., 2021; ElSayed-Aly et al., 2021; Pranger et al., 2021). Central to shielding is the notion of a *shield*, a filter that checks actions proposed by the agent's existing policy with reference to a model of the environment's dynamics and some formal safety specification. The shield overrides actions that may lead to an unsafe state using some other safe (but by no means optimal) policy. A key advantage of many shielding approaches is that the resulting *shielded* policies are formally verifiable; however, a shortcoming is that they require a model of environmental dynamics - typically handcrafted - to be provided in advance. Providing such a model may prove difficult for complex real-world environments, with inaccuracies and human biases creeping into handcrafted models.

In this work, we propose a safe RL agent that makes uses of *latent shielding*, an approach to shielding in environments where a formally-specified dynamics model is not available in advance. At an intuitive level, the agent uses a data-driven approach to learn its own latent world model (a component of which is a dynamics model) which is then leveraged by a shield. The shield then uses the agent's model to "imagine" trajectories arising from different actions, forcing the agent to avoid

those it foresees leading to unsafe states. In addition, the agent can be trained within its own latent world model thus reducing the number of safety violations seen during training.

**Contributions** The main contribution of this work is a framework for shielding agents in complex, stochastic and high-dimensional environments without knowledge of environmental dynamics *a priori*. We further introduce a new method to aid exploration when training shielded agents. Though our framework loses the formal safety guarantees associated with traditional symbolic shielding approaches, our experiments illustrate that latent shielding reduces unsafe behaviour during training and achieves testing performance comparable to previous symbolic approaches.

## 2 PRELIMINARIES

In this section, we cover some relevant background topics. We begin by introducing our problem setup for safety-aware RL and give an overview of the specification language used in this work. This is followed by an outline of the latent world model we make use of in this work as well as a discussion on shielding.

### 2.1 PROBLEM SETUP

We consider an agent interacting with an environment $\mathcal{E}$ modelled as a *partially observable Markov decision process (POMDP)* with states $s \in \mathcal{S}_{\mathcal{E}}$, observations $o_t \in \mathcal{O}_{\mathcal{E}}$, agent-generated actions $a_t \in \mathcal{A}_{\mathcal{E}}$ and scalar rewards $r_t \in \mathbb{R}$ over discrete time steps $t \in [0, 1, ..., T-1]$. We assume the environment has been augmented with a labelling function $L_{\mathcal{E}}^{\phi} : \mathcal{S}_{\mathcal{E}} \to \{safe, unsafe\}$ that, at each time step, informs us whether a *violation* has occurred with respect to some formal safety specification $\phi$. For the avoidance of doubt, we define a violation to have occurred whenever $\phi$ does not hold. This is a weaker assumption than previous works in shielding (which assume access to an abstraction of the environment) and can be thought as a secondary safety-focused reward function with a binary output. Intuitively, the goal of the agent is to learn a policy $\pi$ that maximises its expected cumulative reward while minimising the number of violations of $\phi$.

### 2.2 SYNTACTICALLY CO-SAFE LINEAR TEMPORAL LOGIC

In this work, we use *syntactically co-safe Linear Temporal Logic (scLTL)* (Kupferman & Vardi, 2001) as our specification language. Valid scLTL formulae over some set of atomic propositions $AP$ can be constructed according to the following grammar:

$$\phi ::= true \mid d \mid \neg d \mid \phi \vee \phi \mid \phi \wedge \phi \mid \bigcirc \phi \mid \phi \cup \phi \mid \diamond \phi \tag{1}$$

where $d \in AP$, $\neg$ *(negation)*, $\vee$ *(disjunction)*, $\wedge$ *(conjunction)* are the familiar operators from propositional logic, and $\bigcirc$ *(next)*, $\cup$ *(until)* and $\diamond$ *(eventually)* are temporal operators. We can monitor a co-safe LTL specification using a technique known as *progression* (Bacchus & Kabanza, 2000).

### 2.3 RECURRENT STATE-SPACE MODELS

We refer to the predictive model of an environment maintained by a model-based agent as its *world model*. World models can be learnt from experience and be used both as a substitute for the environment during training (Ha & Schmidhuber, 2018; Hafner et al., 2021) and for planning at run-time (Hafner et al., 2019b). Though many realisations of the notion of a world model exist, the world model used in this work is based on the *recurrent state-space model (RSSM)* proposed by Hafner et al. (2019b).

An RSSM is composed of three key components: a latent dynamics model, a reward model, and an observation model. These components act on compact states formed from the concatenation of a deterministic latent state $h_t$ and stochastic latent state $z_t$.

**Latent Dynamics Model** The latent dynamics model is made up of a number of smaller models. First, the *recurrent model* $h_t = f(h_{t-1}, z_{t-1}, a_{t-1})$ is used to compute the deterministic latent state based on the previous compact state and action. From $h_t$ and the current observation $o_t$, a

distribution $q(z_t|h_t, o_t)$ over posterior stochastic latent states $z_t$ is computed by the *representation model*. At the same time, a distribution $p(\hat{z}_t|h_t)$ over prior stochastic latent states $\hat{z}_t$ is computed by the *transition model*, based only on $h_t$. During training, the transition model attempts to minimise the *Kullback Leibler (KL)* divergence between the prior and posterior stochastic latent state distributions. In doing this, the RSSM learns to predict future latent states (using the recurrent and transition models) without access to future observations.

**Observation Model**   The *observation model* computes the distribution $p(\hat{o}_t|h_t, z_t)$ over observations $\hat{o}_t$ for a particular state. Though not strictly needed, the observation model can prove useful for visualising predicted future states and providing a richer training signal.

**Reward Model**   The *reward model* computes the distribution $p(\hat{r}_t|h_t, z_t)$ over rewards $\hat{r}_t$ for a particular state.

In practice, the distributions $p$ and $q$ are implemented with neural networks $p_\theta$ and $q_\theta$ respectively, parameterised by some set of parameters $\theta$. These latent dynamics models define a fully-observable *Markov decision process (MDP)* as the latent states in the agent's own internal model can always be observed by the agent (Hafner et al., 2019a). We denote the state space of this MDP (comprised of compact latent states) as $\mathcal{S}_\mathcal{I}$.

## 2.4   SHIELDING

The *classical* formulation of shielding in RL is given by Alshiekh et al. (2018). It assumes access to two ingredients: an LTL safety specification and *abstraction* (a MDP model of the environment that captures the aspects of the environment relevant for planning ahead with respect to the safety specification). These ingredients are used to construct a formally verifiable reactive system that monitors the agent's actions, overriding those which lead to violation states.

Proposed by Giacobbe et al. (2021), *bounded prescience shielding (BPS)* avoids the need for hand-crafted abstractions by exploiting the fact that some agents are trained in computer simulations. The shield operates by leveraging access to the program underlying the simulation to look ahead into future states within some finite horizon. Using BPS over classical shielding does, however, come with a few disadvantages. Firstly, it requires access to the simulation at run-time which may prove difficult to provide (especially in cases where running the simulation is computationally expensive). Moreover, an agent using BPS, even when starting from a safe state, can find itself entering unsafe states in cases where the number of steps between a violation being caused by an action and the violation state itself exceeds the shield's look-ahead horizon. This is not the case for classical shielding which resembles BPS with an infinite horizon.

## 2.5   BOUNDED SAFETY

The notion of safety used by BPS is defined over MDPs. For an arbitrary MDP with states $\mathcal{S}$ and actions $\mathcal{A}$, a *bounded trajectory* $\rho$ of length $H$ is a sequence of states and actions $s_0 \xrightarrow{a_0} s_1 \xrightarrow{a_1} \ldots \xrightarrow{a_{n-1}} s_n$ comprised of no more than $H$ states and with the final state $s_n$ either being a terminal state or $n = H - 1$. We further denote the set of all finite trajectories starting from some arbitrary state $s \in \mathcal{S}$ by $\varrho(s)$ and the set of all bounded trajectories of length $H$ that start from $s$ by $\varrho_H(s)$.

We say a bounded trajectory $\rho$ of length $H$ satisfies $H$-*bounded safety* with respect to safety specification $\phi$, written $S_H(\rho, \phi)$, if and only if for all $s_i \in \rho$, $L_\mathcal{E}^\phi(s_i) = safe$. Moreover, we can extend the notion of $H$-bounded safety over the set of policies: a policy $\pi$ is $H$-*bounded safe* with respect to $\phi$, denoted as $S_H(\pi, \phi)$, if and only if for all s $\in \mathcal{S}$,

- either there exists some $\rho \in \varrho_H(s)$ such that $S_H(\rho, \phi)$ and $\pi(s_0) = a_0$;

- or for all $\rho \in \varrho_H(s)$, $\neg S_H(\rho, \phi)$.

In other words, the policy will choose a safe trajectory as long as one exists. Finally, we formally define a violation of $\phi$ to be *inevitable* in state $s_0 \in \mathcal{S}$ if and only if for all $\rho \in \varrho(s_0)$, $\neg S_H(\rho, \phi)$.

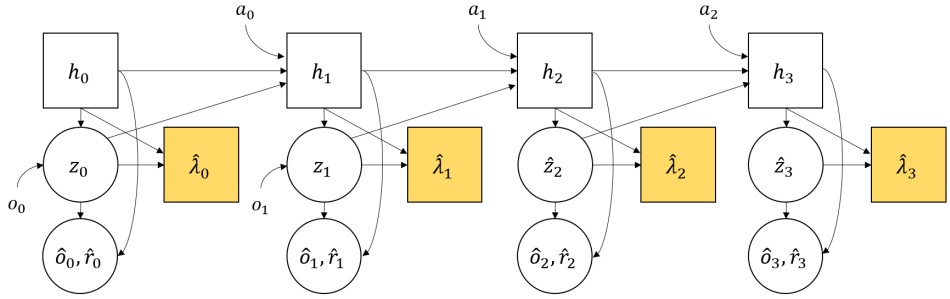

Figure 1: Safety RSSM. In this example, the model observes the environment for two time steps and predicts the subsequent two time steps. $a_t$ denotes the action taken at step $t$, $o_t$ the observation, $h_t$ the deterministic component of the latent state, $z_t$ the posterior stochastic component, $\hat{z}_t$ the prior stochastic component, $\hat{\lambda}_t$ the violation prediction (our novel contribution, highlighted in yellow), $\hat{o}_t$ the predicted observation and $\hat{r}_t$ the predicted reward. Circles represent stochastic variables whereas squares represent deterministic variables. An arrow from one shape to another indicates that the source is used in the calculation of the destination.

## 3 SHIELDED DREAMS

We introduce the notion of *latent shielding*, a novel class of shielding approaches that replace abstractions used by shields with learned latent dynamics models thus allowing the enforcement of formal safety specifications while avoiding the need for an explicitly-defined abstraction of the environment. We further introduce the first such approach, *approximate bounded prescience shielding*, a framework for latent shielding that leverages latent world models learnt by model-based *deep RL (DRL)* agents. In this work, our model-based agent of choice is Dreamer (Hafner et al., 2021)[1], which we modify to incorporate shielding into its data collection, training and deployment phases.

### 3.1 SAFETY RSSM

We augment the standard RSSM with a labelling function $L_{\mathcal{I}}^{\phi} : \mathcal{S}_{\mathcal{I}} \to \{safe, unsafe\}$ which maps latent states $s_t \in \mathcal{S}_{\mathcal{I}}$ to whether they correspond to states in violation of $\phi$. As with the other models, $L_{\mathcal{I}}^{\phi}$ is implemented with a neural network with a categorical output $l_{\theta}$ also parameterised by $\theta$. This yields an enhanced RSSM (illustrated in Figure 1) which we will refer to as a *safety RSSM (SRSSM)*.

We train $l_{\theta}$ along with the other components of the SRSSM with the objective

$$\min_{\theta} \mathcal{L}_{model} = \mathcal{L}_{observation} + \mathcal{L}_{reward} + \mathcal{L}_{KL} + \mathcal{L}_{violation} \tag{2}$$

where the first three terms are as described in (Hafner et al., 2019a; 2021) and, for convenience, can be found in Appendix B. $\mathcal{L}_{violation}$ is a new term that we introduce that acts as a weighted binary cross-entropy loss over predictions by $l_{\theta}$

$$\mathcal{L}_{violation} = -\frac{1}{|\mathcal{D}_{\lambda}|} \sum_{((h,z),\lambda) \in \mathcal{D}_{\lambda}} (\alpha\lambda \log(l_{\theta}(h,z)) + (1-\alpha)(1-\lambda)\log(1 - l_{\theta}(h,z))) \tag{3}$$

where $\mathcal{D}_{\lambda} \subseteq \mathcal{S} \times \{0,1\}$ is a dataset of state-label pairs (with 0 corresponding to $safe$ and 1 to $unsafe$) and $\alpha \in [0,1]$ is a weighting term. The inclusion of the weighting hyperparameter $\alpha$ is motivated by the observation that the number of observations of safe and unsafe states will generally be quite imbalanced. Thus, to prevent predictions by $l_{\theta}$ from collapsing to a single class, we assign a greater weight to the less-represented class.

### 3.2 APPROXIMATE BOUNDED PRESCIENCE SHIELDING

We now integrate the SRSSM as part of a latent shielding approach which we shall refer to as *approximate bounded prescience shielding (ABPS)*. Though ABPS is inspired by BPS, it differs in

---

[1]In practice, any model-based agent with a latent dynamics model can be used.

two key aspects: (1) we approximate the labelling function $L_{\mathcal{E}}^{\phi}$ and environmental dynamics using an SRSSM; and (2) we sample a fixed number of potential future trajectories as opposed to exhaustively exploring all possibilities.

Thus, our approach can be thought of as an approximation of some "ideal" bounded prescience shield that uses the true environmental dynamics and labelling function. The advantage of the first difference should be obvious: it enables the shield to learn its own abstraction, removing the need for hand-crafting or access to a digital environment's underlying program. Why the second difference is advantageous may be slightly less obvious - it's a heuristic that allows us to increase the horizon $H$. By directing the sampling of trajectories in accordance with states and actions the policy is biased towards (as opposed to uniformly), it may be possible to achieve comparable performance to a standard BPS in foreseeing unsafe states. The intuition behind this is that by not sampling trajectories the agent is unlikely to take, more of our computational budget can be dedicated into ensuring that the agent's most likely trajectories are safe. In other words, we will not spend time planning to correct actions that the agent is unlikely to take.

More formally, our shielded policy $\pi^*$ can be written:

$$\pi^*(s_t) = \begin{cases} \pi(s_t), \text{ if } P(l_\theta(s_{t+1}) = 1|\pi(s_t)) < \epsilon \\ \pi_{alt}(s_t), \text{ otherwise} \end{cases} \tag{4}$$

where $\pi(s_t)$ is the agent's policy without shielding; $s_t \in \mathcal{S}_{\mathcal{I}}$ is some compact latent state; and $\pi_{alt}$ is an alternative policy that ensures that, if a violation isn't already inevitable, the agent avoids a predicted unsafe state. In our implementation, $\pi_{alt}$ simply considers all the other actions until a safe trajectory is found. In the event that no safe action is found, the agent takes a random action.

Since the SRSSM models stochasticity, we can sample multiple futures arising from an action being taken and derive probabilistic estimates of whether an action will lead to a violation. In this way, we can estimate the safety of an action taken in a given state by checking whether the probability of a violation occurring (inferred by sampling) exceeds a fixed threshold $\epsilon$ (see Algorithm 1). Moreover, the sampling process can, in practice, be augmented to sample a wider range of trajectories (less likely to be taken by the agent) by adding a small amount of noise to actions suggested by the policy during sampling.

---

**Algorithm 1:** Approximate Bounded Prescience Shielding in latent space.

**Input:** Current compact latent state $(h, z)$, unshielded action $a$, horizon $H$, number of
        trajectories to sample $N$, alternative policy $\pi_{alt}$ and threshold $\epsilon$.
**Output:** A tuple containing an action and whether or not the shield had to interfere.

1   Initialise array of zeros $\Lambda$ with length $N$.
2   **for** $n = 1..N$ **do**
3      Imagine trajectory $\rho = (h, z) \xrightarrow{a} (\hat{h}_1, \hat{z}_1) \xrightarrow{a_1} ... \xrightarrow{a_{H-1}} (\hat{h}_H, \hat{z}_H)$ where $a_i = \pi(\hat{h}_i, \hat{z}_i)$.
4      **if** $\exists i \in [1, 2, ..., H]$ such that $l_\theta(h_i, z_i) = unsafe$ **then** $\Lambda[n] := 1$.
5   **end**
6   $interfered := false$.
7   **if** $\frac{1}{N} \sum_{\lambda \in \Lambda} \lambda > \epsilon$ **then**
8      $a, interfered := \pi_{alt}(h_i, z_i), \; true$.
9   **end**
10   **return** $\langle a, interfered \rangle$.

---

### 3.3 TRAINING REGIME

We extend the training regime proposed in Hafner et al. (2019a) to include the training and application of ABPS. Though the full training procedure is detailed in Algorithm 2, we provide an overview below.

### 3.3.1 Overview

The training procedure can be split into three phases: data collection, latent world model training and agent training (lines 10-22, 4-6 and 7-8 in Algorithm 2 respectively). These phases are cycled through until convergence.

**Data Collection**  In this phase, the agent interacts with the real environment to collect a dataset $\mathcal{D} \subseteq \mathcal{S} \times \mathcal{A} \times \mathcal{O} \times \mathbb{R} \times \{0, 1\}$ of states, actions, observations, rewards and violations with which a latent world model can be learned. At the very start of training, we collect trajectories from $S$ seed episodes using a random policy; at all other times, we use the agent's shielded policy. An illustration of the components of the agent active during this phase can be found in Figure 4 in Appendix A.

**Latent World Model Training**  The goal of this phase is to improve the model of the world so that the policy has an accurate imagined environment to train in. To this end, we train the latent world model with respect to the objective in Equation 2 on $B$ data sequences of length $L$ sampled from $\mathcal{D}$. An illustration of the components of the agent active during this phase can be found in Figure 5 in Appendix A.

**Agent Training**  Agent training is composed of two steps. First, starting from states from the same $B$ data sequences from the latent world model training phase, the agent imagines $I$ trajectories of length $H$ with actions chosen from its current policy. Next, the unshielded policy $\pi$ is updated as in Hafner et al. (2019a; 2021).

### 3.3.2 Key Changes and Contributions

We describe and discuss the key changes we have made that differentiate our agent from previous approaches.

**Experience Dataset.** Elements in the experience dataset $\mathcal{D}$ now contain a binary variable representing whether a violation has occurred $\lambda_t$.

**Latent Shielding.** Before being sent to the environment $\mathcal{E}$, actions $a_t$ generated by the unshielded policy $\pi$ are routed through our newly-proposed latent shield (described in Algorithm 1). The shield returns a new approximately $H$-bounded safe action $a_t'$.

**Intrinsic Punishment.** Whenever a violation occurs (whether detected by the latent shield or by the environment), we override the environment's reward function by instead assigning a reward $p < 0$. This discourages the agent's unshielded policy $\pi$ from taking actions that lead to unsafe states through the standard RL setup. Over time, this means that the shield will have to interfere less as $\pi$ becomes biased away from unsafe states. This is a standard practice in the shielding literature (Alshiekh et al., 2018), however new to the Dreamer family of agents (Hafner et al., 2019a;b; 2021).

**Shield Introduction Schedule**  We introduce the novel notion of a *shield introduction schedule* $\Pi$ which can enable and disable shielding during training. The rationale behind this is that a shield backed by an inaccurate world model will incorrectly label some safe states as unsafe, and vice versa. In some cases, this may prove detrimental to the training process: suppose $l_\theta$ happens to be initialised in such a way that all states are labelled as unsafe. As a result, the shield can prevent the agent from exploring the environment and improving its internal model of the environment. This, in turn, can prevent the labelling function from learning to correctly differentiate safe states from unsafe states.

To give the agent time to learn a good world model before restricting exploration through a learned shield, we augment the training procedure with $\Pi$ which aims to gradually introduce shielding. To our knowledge, this is the first time such a system has been proposed. Though there exist a wide range of possible implementations of $\Pi$, in this work we use a simple schedule that seems to work well empirically: start training with shielding disabled and enable shielding once the world model loss begins to plateau (see Appendix H for more details). A detailed comparison of different shield introduction schedules, however, is beyond the scope of this study and left for further work.

---

**Algorithm 2:** Training Dreamer with Approximate Bounded Prescience Shielding.

---

**Input:** Punishment for violation $r_{punish}$, shield imagination horizon $H$, number of steps to imagine ahead for training $I$, number of seed episodes $S$, number of training steps per episode $C$, number of real environment interaction steps per episode $T$, sequence length $L$, batch size $B$, environment $\mathcal{E}$, labelling function $L_{\mathcal{E}}^{\phi}$ and shield introduction schedule $\Pi$.

1   Initialise dataset $\mathcal{D}$ with $S$ seed episodes and neural network parameters $\theta$ randomly.
2   **while** *not converged* **do**
     `/* Learning the internal world model and agent policy.      */`
3     **for** *c = 1..C* **do**
4        Draw $B$ data sequences $\{(a_t, o_t, r_t, \lambda_t)_{t=k}^{k+L}\} \sim \mathcal{D}$.
5        Compute model states $h_t$, $z_t$ and $\hat{z}_t$.
6        Update $\theta$ using representation learning.
7        Imagine a trajectory $\rho$ of length $I$ for each state in each data sequence.
8        Update the unshielded policy $\pi$ based on the imagined trajectories.
9     **end**
     `/* Collecting data from the environment.                     */`
10    **for** *t = 1..T* **do**
11       Compute model states $h_t$, $z_t$ from history.
12       Select action $a_t$ with the unshielded policy, adding exploration noise if desired.
13       **if** $\Pi$ *has enabled shielding* **then**
14         Pass $h_t$, $z_t$ and $a_t$ into Algorithm 1 to obtain the tuple $\langle a_t', interfered \rangle$.
15       **else**
16         $a_t', interfered := a_t, false$.
17       **end**
18       $r_t, o_t := \mathcal{E}.\texttt{step}(a_t')$.
19       Check if $L_{\mathcal{E}}^{\phi}$ has detected a violation and store the result as 0 or 1 in $\lambda_t$.
20       **if** $\lambda_t = 1$ or *interfered* **then** $r_t := r_{punish}$.
21     **end**
22    Add experience to dataset $\mathcal{D} := \mathcal{D} \cup \{(o_t, a_t, r_t, \lambda_t)_{t=1}^{T}\}$.
23   **end**

---

## 4   EXPERIMENTS

In this section, we compare our ABPS agent against Dreamer without shielding (Hafner et al., 2019a; 2021), Dreamer with BPS (Giacobbe et al., 2021), and CPO (Achiam et al., 2017). We also empirically investigate some aspects of the internal workings of our agent. A brief overview of the environments used can be found below with more detailed a discussion in Appendix C.

**Visual Grid World**   The *Visual Grid World (VGW)* environment is a simple deterministic navigation benchmark with high-dimensional ($64 \times 64 \times 3$) visual observations (see Figure 2) and discrete actions. The agent's (in green) task is to navigate to randomly-placed targets (in black) while avoiding any unsafe locations (in red). We experiment with both *fixed* and *procedurally generated* grids.

**Cliff Driver**   The *Cliff Driver (CD)* environment is a symbolic benchmark with stochastic dynamics and continuous actions. The agent controls the forward acceleration of a car and is tasked with driving to the edge of a cliff as quickly as possible without overshooting and falling into the sea. Stochasticity comes from the fact that at each time step there is a probability $p_{stick}$ that car's controls become "stuck" (Machado et al., 2018).

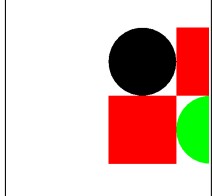

### 4.1   PERFORMANCE EVALUATION

We evaluate our agents on three metrics: (1) average reward per episode at test-time; (2) average number of violations per episode at test-time; and (3)

Figure 2: Sample visual observation from the VGW environment.

| | Flavour | Metric | Latent | Unshielded | BPS | CPO |
|---|---|---|---|---|---|---|
| Visual Grid World | Fixed | Testing Reward | **15067 (434)** | 13148 (249) | 12468 (620) | -2925 (1065) |
| | | Testing Violations | 0.30 (0.76) | 2.25 (1.60) | **0 (0)** | 13.43 (19.25) |
| | | Training Violations | 1262 (172) | 2306 (833) | **0 (0)** | 16455 (1435) |
| | Procedural | Testing Reward | **8084 (2221)** | 6825 (1427) | 1938 (3552) | -1588 (2051) |
| | | Testing Violations | 4.50 (3.59) | 33.7 (16.28) | **0 (0)** | 19.60 (13.83) |
| | | Training Violations | 14018 (1852) | 15309 (4686) | **0 (0)** | 18705 (3756) |
| Cliff Driver | $p_{stick} = 0.1$ | Testing Reward | 8.57 (2.96) | **10.76 (3.29)** | 10.50 (3.28) | 7.56 (2.86) |
| | | Testing Violations | **0 (0)** | **0 (0)** | **0 (0)** | 3.40 (1.91) |
| | | Training Violations | 58.2 (9.60) | 90.0 (9.10) | **24.0 (13.02)** | 973.0 (357.7) |
| | $p_{stick} = 0.5$ | Testing Reward | **8.10 (4.99)** | 6.63 (8.07) | 7.10 (9.52) | 6.44 (3.00) |
| | | Testing Violations | **0.18 (0.84)** | 0.54 (1.53) | 0.22 (1.18) | 0.48 (1.24) |
| | | Training Violations | 91.8 (16.85) | 157.6 (18.4) | **80.4 (17.43)** | 3126 (2823) |

Table 1: Comparison of trained agents with standard deviations in parentheses where applicable.

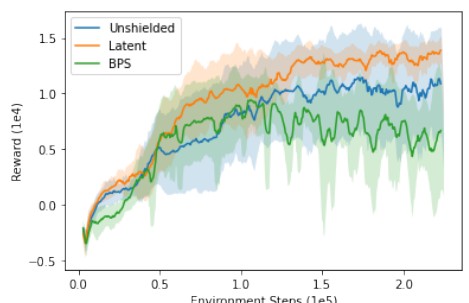 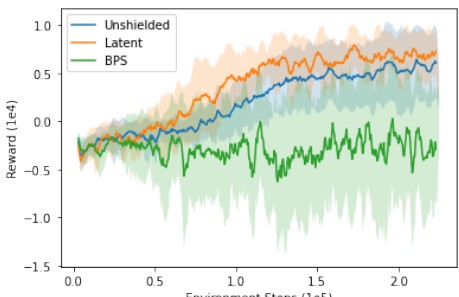

Figure 3: Reward curves for the model-based agents on fixed (left) and procedurally generated (right) instances of the VGW environment. The shaded area denotes one standard deviation.

total number of violations during training. Results are calculated by averaging over five trained versions of each agent. The latent shield horizon $H$ was 2 and 6 for the VGW and CD environments respectively. Moreover, since the action space of the CD environment is continuous, we discretise the actions into four bins when performing BPS (see Appendix D.5). Such modifications, however, are not needed for our agent. Full details on training and evaluation can be found in Appendix D. We also provide further evaluation of our agents on environment-specific metrics in Appendix F.

As can be seen in Table 4, our agent collected more reward at test-time than both the unshielded and BPS agents in the VGW environment (reward curves during training can be seen in Figure 3 with further training curves showcased in Appendix E). Moreover, our agent saw a seven-fold reduction in test-time violations compared to the unshielded agent across both the fixed and procedurally generated environments, averaging at less than one violation per episode for the fixed VGW. This reduction in training violations is even more dramatic when compared to the model-free CPO agent. Our agent outperformed the other agents at test-time in the most stochastic CD environment ($p_{stick} = 0.5$). This is possibly due to the SRSSM used by our latent shield being better able to capture non-determinism than BPS. Nevertheless, the result is still rather impressive given that the agent using BPS sampled 1024 trajectories at every time step whereas our agent only sampled 20.

## 4.2 OTHER EXPERIMENTS

**Qualitative Evaluation of Learned Latent Dynamics** We evaluate the quality of each agent's latent dynamics model by observing its trajectory predictions given a particular starting state and action sequence. Each agent is given the same starting observation and predefined sequence of 10 actions. The agents take these actions in their respective latent world models with the states traversed being decoded for inspection. Results can be seen in Appendix G and indicate that, qualitatively, our agent's decoder performed the best, accurately predicting 10 frames into the future. One possible reason for this observation is that the inclusion of the violation detection objective in the SRSSM encourages the model to focus on accurately capturing violation states. Another potential factor is that the BPS agent never actually enters unsafe states and thus finds it difficult to represent them.

**Do Shield Introduction Schedules Actually Help?** We perform an ablation study with shield introduction schedules in the VGW environment. Results can be found in Appendix I and demonstrate decreased performance in the absence of shield introduction schedules.

## 5 RELATED WORK AND DISCUSSION

**Latent World Models** Learning latent world models from visual observations has seen growing interest from the RL community (Wahlström et al., 2015; Watter et al., 2015; Racanière et al., 2017; Ha & Schmidhuber, 2018; Hafner et al., 2019a;b; Schrittwieser et al., 2020; Hafner et al., 2021). One key trend in the literature is that of training agents in their own learned world models (Ha & Schmidhuber, 2018; Hafner et al., 2019a; 2021). In this work, we extend the world model formulation used in Hafner et al. (2019a;b; 2021) to encode a notion of safety into state representations.

**Safety By Filtering Actions** Overriding unsafe actions with safe ones has been a popular approach to safety-aware RL. Introduced into the RL scene by Alshiekh et al. (2018), shielding has received much research interest and has seen applications in real-world settings (Nikou et al., 2021). Various works have attempted to address some of its shortcomings of the original formulation. These include allowing the shield to be updated to aid exploration or correct model imperfections (Anderson et al., 2020; Pranger et al., 2021); improving performance in non-deterministic environments (Jansen et al., 2020; Li & Bastani, 2020); and extending shielding to multi-agent RL (ElSayed-Aly et al., 2021). To our knowledge, few works (Srinivasan et al., 2020; Thananjeyan et al., 2021; Bharadhwaj et al., 2021; Giacobbe et al., 2021) focus on removing the need for handcrafting an abstraction (among the most time-consuming and error-prone aspects of shielding), and only one of them (Giacobbe et al., 2021) achieves this without getting rid of the abstraction altogether, albeit by providing the agent with access to the program that controls the environment.

In contrast, latent shielding tackles the problem head-on by directly learning an abstraction for use by the shield. In this work, the abstraction we use is an SRSSM, which captures stochasticity by design, a useful property for non-deterministic environments. Though our latent shield satisfies an approximation of $H$-bounded safety (see Section 2.4) with respect to the learned abstraction, its safety with respect to the true environmental dynamics is not guaranteed and instead relies on the fidelity with which the true dynamics are captured. Furthermore, unlike for its formally-verified predecessors, it is a necessary sacrifice that the agent visits unsafe states during training in order to learn a notion of safety (unless, of course, pre-training is possible). Nevertheless, a learned abstraction may be advantageous in settings where handcrafting an abstraction is not feasible and privileged access to some simulation (as in Giacobbe et al. (2021)) cannot be assumed. Moreover, latent shielding may provide a greater degree of explainability over model-free methods which get rid of the abstraction altogether (Srinivasan et al., 2020; Thananjeyan et al., 2021; Bharadhwaj et al., 2021) - if the shield overrides an action, one can reconstruct the imagined unsafe trajectories that led to the interference.

## 6 CONCLUSIONS

In this paper, we have presented *latent shielding*, a new framework for shielding DRL agents without the need for a handcrafted abstraction of the environment. Using this framework, we have designed a novel shield and demonstrated that this method not only leads to improved adherence to safety specifications on two benchmark environments with respect to an unshielded agent, but also works out-of-the-box on both continuous and discrete environments (unlike its predecessor, BPS). Furthermore, we have demonstrated for the first time that shielding at inappropriate times may adversely impact the performance of model-based DRL agents and showed how this phenomenon can be counteracted using our novel notion of *shield introduction schedules*.

Our work opens several exciting avenues for future work. For instance, this work uses a very simple shield introduction schedule; future work may provide a richer investigation into the properties of different schedules. Moreover, though demonstrating promising empirical results, our realisation of latent shielding loses the formally verifiable safety guarantees enjoyed by many symbolic shielding approaches - whether it is possible to construct a verifiable implementation of latent shielding, or compensate for the loss of formal guarantees, are open problems.

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

## A    FURTHER HELPFUL DIAGRAMS

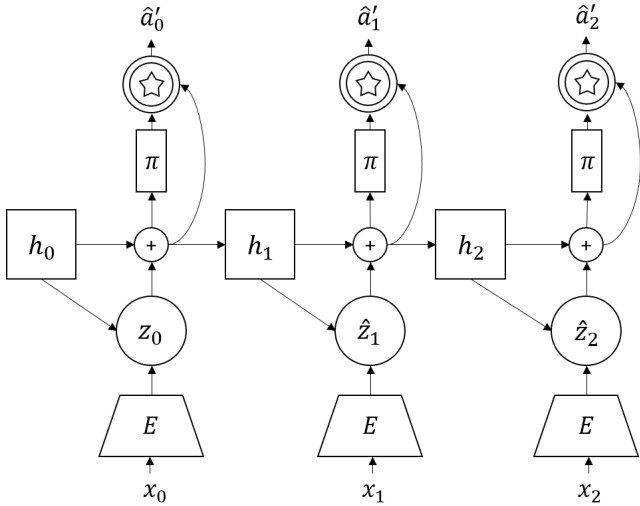

Figure 4: Interacting with the real environment. Observations $x_t$ from the environment are encoded and a latent representation is obtained. The representation is passed to $\pi$ and the shield to obtain an H-bounded safe (with respect to the agent's world model) action $\hat{a}'_t$. The circle with the plus sign represents the concatenation operation.

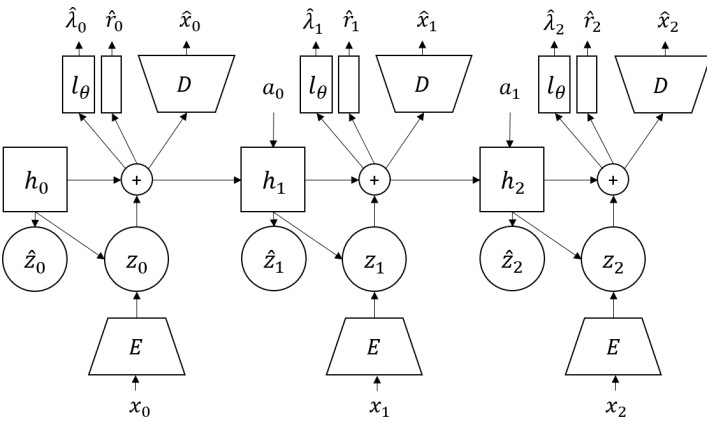

Figure 5: Training a world model. Observations $x_t$ from the experience dataset $\mathcal{D}$ are encoded by $E$ and used to generate a posterior stochastic state $z_i$. $z_i$ is concatenated with the deterministic state $h_i$ and used to predict the reward $\hat{r}_i$, original observation $\hat{x}_i$ (through decoder $D$), and whether or not a violation has taken place $\hat{\lambda}_i$ (though network $l_\theta$). A prior stochastic state $\hat{z}_i$ is also generated with a view to matching the posterior state as closely as possible. The circle with the plus sign represents the concatenation operation.

# B  DREAMER LOSS FUNCTIONS

Given some dataset of experience $\mathcal{D}_e \subseteq \mathcal{S} \times \mathcal{O}_\mathcal{E} \times \mathbb{R}$:

$$
\begin{aligned}
\mathcal{L}_{observation} &= -\frac{1}{|\mathcal{D}_e|} \sum_{((h,z),o,r)\in\mathcal{D}_e} \log p(o|h,z) \\
\mathcal{L}_{reward} &= -\frac{1}{|\mathcal{D}_e|} \sum_{((h,z),o,r)\in\mathcal{D}_e} \log p(r|h,z) \\
\mathcal{L}_{KL} &= \frac{1}{|\mathcal{D}_e|} \sum_{((h,z),o,r)\in\mathcal{D}_e} \beta D_{\mathrm{KL}}(z\|\hat{z}) + (1-\beta)D_{\mathrm{KL}}(\hat{z}\|z)
\end{aligned}
\tag{5}
$$

where $\beta$ is a hyperparameter for *KL balancing* (Hafner et al., 2021). Here, minimising $\mathcal{L}_{observation}$ and $\mathcal{L}_{reward}$ corresponds to maximising the log-likelihood of $\hat{o}$ and $\hat{r}$ respectively; minimising $\mathcal{L}_{KL}$ corresponds to minimising the KL divergence between the $z$ and $\hat{z}$.

# C  ENVIRONMENT DETAILS

## C.1  VISUAL GRID WORLD

The *Visual Grid World* is a simple visual navigation benchmark. The environment (see Figure 2) is made up of the agent (green circle), a target (black circle) and a number of unsafe locations (red squares). The actions the agent can take correspond to moving up, down, left, right and staying still; visual observations from the environment are encoded in $64 \times 64 \times 3$ arrays. The agent's task is to navigate to the target, avoiding any unsafe locations. This yields a relatively straightforward safety specification:

¬agent_in_red_square ∪ episode_ended

where `agent_in_red_square` is true if and only if the agent is in an unsafe location, and `episode_ended` is only true at the end of an episode.

The reward function (see Equation 6) used is also quite simple, with a small penalty term at each time step to encourage movement (though indeed different forms of intrinsic motivation may be also used).

$$
\mathcal{R}(s_t, a_t) = \begin{cases}
100, & \text{agent reaches target,} \\
-40, & \text{agent enters unsafe state,} \\
-10, & \text{agent does not move,} \\
-1, & \text{otherwise}
\end{cases}
\tag{6}
$$

We train and test our models on randomised 4x4 instances of the environment, each with either one or two unsafe states.

## C.2  CLIFF DRIVER

The *Cliff Driver* environment is a simple symbolic benchmark with stochastic dynamics. The agent controls a car on a one-dimensional road with continuous actions $a_t \in [-1, 1]$ which correspond to the acceleration of the car. The car cannot move backwards and its speed $v_t$ is lower-bounded at 0 (thus $a_t < 0$ corresponds to braking as opposed to reversing). At the end of the road, there is a cliff. The agent starts each episode stationary at a fixed distance $x_0$ from the edge of the cliff with its distance at subsequent time steps being written $x_t$. Furthermore, at each time step, there is a probability $p_{stick}$ that the car's controls get "stuck" meaning that the current action is ignored and replaced with the previous action.

The agent's goal is to drive the car to the edge of the cliff as quickly as possible, without overshooting and falling into the sea. Observations from the environment are given as two-dimensional vectors encoding the distance from the edge of the cliff in one component and the speed of the car in the other. The safety specification can thus be written:

$$\neg \texttt{agent\_fell\_off\_cliff} \cup \texttt{episode\_ended}$$

where `agent_fell_off_cliff` is true if and only if the agent has overshot the cliff, and `episode_ended` is only true at the end of an episode.

Finally, the reward function is given as:

$$\mathcal{R}(s_t, a_t) = \begin{cases} 1 - \frac{x_t}{x_0}, & \text{agent has not fell off cliff,} \\ -5, & \text{otherwise} \end{cases} \tag{7}$$

# D EVALUATION DETAILS

## D.1 GENERAL

Agents were trained from different seeds on a machine with a single NVIDIA RTX 2080 Ti GPU, an AMD Ryzen 7 2700X processor and 64GB of RAM. All agents were trained using the Adam optimiser (see Appendix D.6 for hyperparameters). Each agent was trained five times, from different random seeds (except the CPO agent in the VGW environments, which, due to computational limitations, was only trained from three seeds). For each environment, the model-based agents were trained for the same number of steps; the CPO agent was allowed to train for more steps ($2\times$ longer for the VGW environments and $5\times$ longer for the CD environments).

## D.2 EVALUATION PROTOCOL

Agents trained on the fixed VGW were evaluated over ten 500-step episodes of the same environment configuration they were trained in. Agents trained on the procedurally generated VGW were evaluated over ten random environment configurations.

Agents trained on the CD environments were evaluated over ten episodes, each with a different random seed.

## D.3 NETWORK ARCHITECTURES

We use the network architectures proposed in Hafner et al. (2019a). The number of nodes in each layer varied depending on the environment and can be found in Appendix D.6. The size of the fully-connected layers and output of the observation encoders and decoders can be found under *Observation Embedding Size*; the size of the fully-connected layers in the reward, value, observation and violation models can be found under *NN Hidden Layer Size*. We implement our encoder for symbolic observations from the CD environment as a feed-forward network with three fully-connected hidden layers and ReLU activations. For the CPO agent, we use the same observation encoders and policy networks as mentioned above.

## D.4 SHIELD INTRODUCTION SCHEDULES

For details on how we constructed these schedules, see Appendix H.

**Visual Grid World** Start with shield disabled. After 10 episodes (including the 5 seed episodes), enable shielding every third episode. After 20 episodes, enable shielding every other episode. After 30 episodes, enable shielding all the time. Linearly decay the unsafe threshold $\epsilon$ form 0.5 to 0.125 over the entire course of training.

**Cliff Driver** Start with shield disabled. After 60 episodes (including the 50 seed episodes), enable shielding all the time.

### D.5 Bounded Prescience Shield for Continuous Action Spaces

In our experiments in the CD environment, we make the action space amenable to BPS by discretising actions into four bins. When the agent outputs a continuous action, we allocate the action to a bin by rounding the action to the nearest action in the set $\{-1, -0.1, 0.1, 1\}$. This rounded action is then sent to the shield, which exhaustively searches through the bounded trajectories generated by taking the actions defined in the aforementioned set.

### D.6 Hyperparameters

Hyperparameter values were kept constant for the base Dreamer model in all agents in each environment. For the CPO agent, we use the same hyperparameters as Ray et al. (2019).

| Hyperparameter | Symbol | Value | |
| --- | --- | --- | --- |
| | | Grid World | Cliff Driver |
| Deterministic State Size | - | 200 | 8 |
| Stochastic State Size | - | 30 | 16 |
| NN Hidden Layer Size | - | 200 | 16 |
| Observation Embedding Size | - | 1024 | 32 |
| Discount Factor | $\gamma$ | 0.99 | 0.99 |
| Action Repeat | - | 1 | 1 |
| Seed Episodes | $S$ | 5 | 50 |
| Episode Length | $T$ | 500 | 20 |
| Batch Size | $B$ | 50 | 250 |
| Sequence Length | $L$ | 50 | 10 |
| Training Steps | $C$ | 100 | 100 |
| Exploration Noise Variance | - | 0.3 | 0.3 |
| Imagination Horizon | $I$ | 15 | 15 |
| KL Balancing Ratio | $\beta : (1 - \beta)$ | 4:1 | 4:1 |
| Violation Balancing Ratio | $\alpha : (1 - \alpha)$ | 1:3 | 1:3 |
| Model Learning Rate | - | 1e-3 | 1e-4 |
| Policy Learning Rate | - | 8e-5 | 8e-5 |
| Value Learning Rate | - | 8e-5 | 8e-7 |
| Bit Depth | - | 5 | - |
| Adam Epsilon | - | 1e-7 | 1e-7 |
| Adam Beta | - | 0.9, 0.999 | 0.9, 0.999 |
| Latent Shield Horizon | $H$ | 2 | 6 |
| Latent Shield Sampled Trajectories | $N$ | 20 | 10 |
| Latent Shield Unsafe Threshold | $\epsilon$ | 0.15 | 0.15 |
| CPO Cost Limit | - | 5 | 0 |

## E Training Curves

Shaded regions in the plots denote a single standard deviation. Note that rewards for the shielded agents may look lower due to intrinsic punishment. That is, the shields can assign punishments (to discourage potentially unsafe behaviour) in addition to the environment's reward function. Furthermore, since the CPO agents were were trained for many more steps than the others, we plot their final performance as single dotted lines.

### E.1 Visual Grid World Reward

See Figure 3.

### E.2 VISUAL GRID WORLD VIOLATIONS

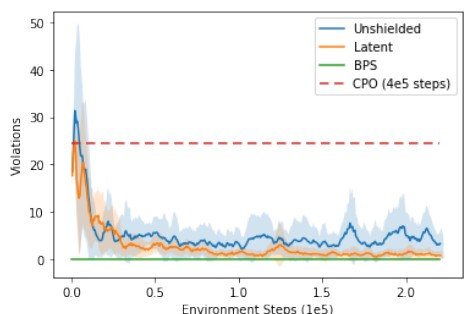 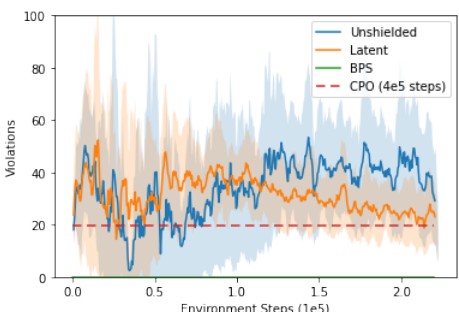

Figure 6: Violations per episode for agents on fixed (left) and procedurally generated (right) instances of the VGW environment.

### E.3 CLIFF DRIVER REWARD

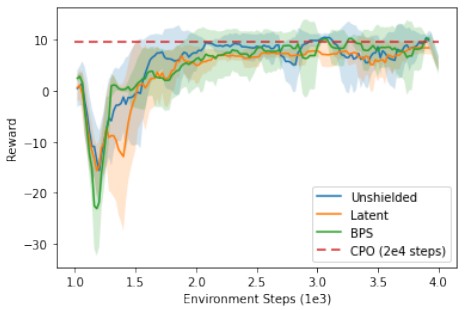 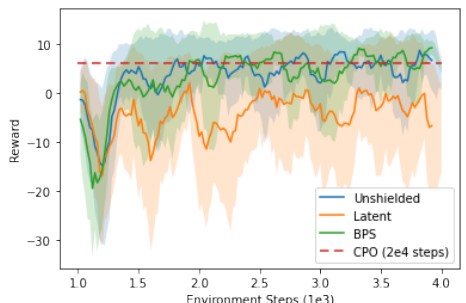

Figure 7: Training reward curves for agents on $p_{sticky} = 0.1$ (left) and $p_{sticky} = 0.5$ (right) instances of the CD environment.

### E.4 CLIFF DRIVER VIOLATIONS

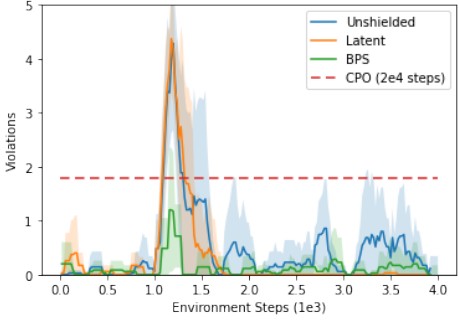 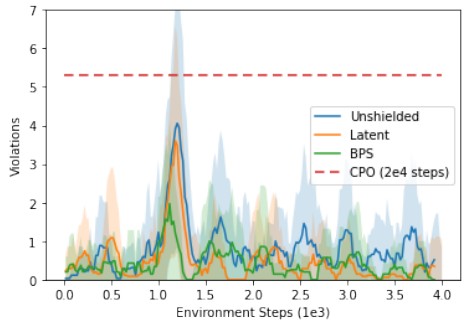

Figure 8: Violations per episode for agents on $p_{sticky} = 0.1$ (left) and $p_{sticky} = 0.5$ (right) instances of the CD environment.

## F ADDITIONAL EVALUATION METRICS

We evaluate the agents on two further environment-specific evaluation metrics below. The first is the average number of targets reached during a VGW episode. The second is the average time taken to travel 90% of the distance to the edge of the cliff during a CD episode.

| | Flavour | Metric | Latent | Unshielded | BPS | CPO |
|---|---|---|---|---|---|---|
| Visual | Fixed | Targets Reached | **154.16 (13.09)** | 136.52 (11.92) | 108.68 (10.74) | 2.53 (2.69) |
| Grid World | Procedural | Targets Reached | 64.36 (32.97) | **84.82 (19.82)** | 55.30 (39.81) | 9.94 (10.37) |
| Cliff | $p_{stick} = 0.1$ | Time Till 90% | **19.4 (4.42)** | 15.96 (4.12) | 15.90 (4.49) | 14.24 (3.79) |
| Driver | $p_{stick} = 0.5$ | Time Till 90% | 18.70 (4.69) | 18.32 (4.88) | **18.78 (4.80)** | 17.38 (4.27) |

Table 2: Comparison of trained agents with standard deviations in parentheses where applicable.

## G  QUALITY OF LATENT DYNAMICS

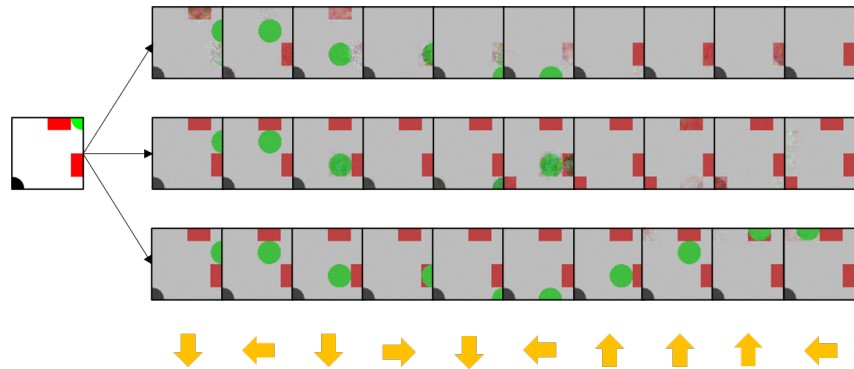

Figure 9: Comparison of latent trajectories in the agents' learned models of the environment. All trajectories begin from the leftmost observation and time progresses from left to right. The bottom row of arrows denote the actions taken just before each time step. From top to bottom, the three rows of images correspond to trajectories in the latent space of the BPS agent, the unshielded agent, and our agent.

## H  METHOD FOR CONSTRUCTING A SHIELD INTRODUCTION SCHEDULE

The method we used to construct our shield introduction schedules is as follows:

1. Train the agent without shielding and plot the violation loss $\mathcal{L}_{violation}$ (see Equation 3).

2. Look for the "elbow" in the plot (see Figure 10 for an example).

3. Disable the shield until around the time of the "elbow". Once the elbow has been reached, enable shielding. If the violation curve does not plateau as nicely as in Figure 10, consider introducing the shield more gradually (e.g. once every $n$ episodes, where $n$ eventually decays to 1).

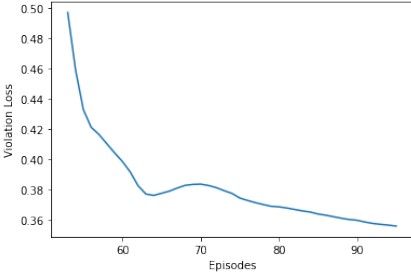

Figure 10: Plot of violation loss $\mathcal{L}_{violation}$ from the CD environment. As you can see, the "elbow" in the curve occurs at around 60 episodes.

# I    ABLATION STUDY WITH SHIELD INTRODUCTION SCHEDULES

We compare the first 100 training episodes of our agent in the fixed VGW environment with and without the shield introduction schedule outlined in Appendix D.4. As in the other experiments, results are calculated over five trained agents. Though both agents start at roughly the same reward, the agents with a shield introduction schedule consistently outperforms their counterparts without shield introduction schedules.

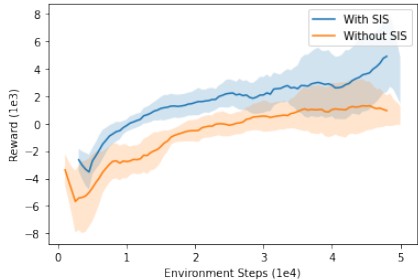

Figure 11: Training reward curves of agents using latent shielding with and without a shield introduction schedule (SIS).

# J    A NOTE ON THE TITLE

Yes, the title of this work is indeed a spin off *Do Androids Dream of Electric Sheep?*. Here, androids dreaming of electric fences refers to RL agents dreaming (i.e. using latent imagination) about being punished for entering forbidden parts of an environment's state space.

