# OpenReview forum: "Do Androids Dream of Electric Fences? Safety-Aware Reinforcement Learning with Latent Shielding"
_ICLR.cc/2022/Conference — ICLR 2022 Submitted_

### Official Review · Reviewer_eXDd · 2021-11-02

**Correctness:** 3
**Technical Novelty And Significance:** 2
**Empirical Novelty And Significance:** 3
**Recommendation:** 5
**Confidence:** 4

**Main Review:**

The proposed approach is simple, and intuitive, which is a major advantage. Fig. 1 clearly describes the contribution of the paper on top of Dreamer, by incorporating a safety constraint violation predictor. The violation predictor is learned based on an approximate shielding approach inspired by classical notions of a shield. The shield checks the probability of constraint violation given the policy outputs a certain action, and if this is the case, then an alternate policy based on rejection sampling is activated. The paper is well-written and easy to follow, with detailed algorithms mentioning the flow of the approach.


I have a few questions and several concerns regarding the paper which I have listed below:

- An important limitation of the approach is that since visiting unsafe states are discouraged during exploration (by penalizing the agent with -negative rewards), the agent is unlikely to visit unsafe states, which is good for being safety but bad for learning an accurate dynamics model in those regions. Hence, dynamics model rollouts are likely to be very inaccurate around unsafe parts of the state-space, thereby making the planning process inaccurate.

- Another limitation of the approach is that it necessarily requires visiting a lot of unsafe states during training, for learning the world model, and hence can never be completely safe. This is also reflected in the results in Table 1 where the training violations are significantly high. It would be helpful to have a discussion about acknowledgement of this limitation in the paper.

- There are no guarantees of how accurate the safety shield is given errors in the learned dynamics model. It will be helpful to formally quantify the safety constraint satisfaction as some function of model-error, or provide intuitions about the extent to which inaccuracies in the model are likely to affect safety violations.

- Since $\pi_{\text{alt}}$ in equation 4 outputs a "safe" action by rejection sampling, there is no guarantee that a safe action if it exists will be found. This is because the learned model is inaccurate, so at any given time, when a safe action is not found, we can't be sure whether this is because a safety violation is inevitable, or because there exists a safe action which is not feasible under the current (inaccurate) dynamics.

-  There are no comparisons to model-free safe RL approaches (for example the constrained optimization and safety critics papers cited in the 2nd paragraph of the introduction). In comparison to prior safe RL works that learn a safe policy without learning a safe dynamics+rewards model i.e. in comparison to safe model-free approaches, the proposed approach will likely suffer from a compounding of errors. This is likely to happen because the proposed method has a large number of learned components (parts of the model) and relies on the model being accurate for planning. Hence, a comparison with some prior model-free safe RL methods is important.

- A very important limitation is the lack of sufficient experiments. Experiments are only on toy settings. It is necessary to evaluate safe behavior on more complex dynamical systems (for example robot manipulation, locomotion, navigation environments). Many of the prior safe RL papers cited evaluate on such environments.

- [clarification] Is the unshielded variant (in blue in the plots), normal Dreamer? If so, then how does the proposed method compare with Dreamer in the normal DeepMind Control Suite environments? It is necessary to see this in order to understand whether the proposed method achieves safe behavior while compromising on task performance in environments where safety is not explicitly important.

- [minor] I am curious if the title has any relevance to the contents of the paper? I understand that it is a spin-off of "Do Androids Dream of Electric Sheep" and the connection to the paper in my understanding is: Dream --> Dreamer; and Fences --> Shield. If this is the case / there is more to it, then a one line explanation of it somewhere in the paper / in the appendix might be helpful.

**Summary Of The Paper:**

This paper proposes a framework for safe model-based RL through latent shielding. The key contribution is extension of an existing MBRL algorithm Dreamer with shielding such that the agent is penalized for taking unsafe actions during exploration; and during planning, the agent can sample multiple plausible futures through the learned dynamics model and avoid taking actions that lead to unsafe states.

**Summary Of The Review:**

This paper modifies a prior model-based RL framework, Dreamer to incorporate a safety violation predictor based on latent shielding. While this idea is simple, easy to implement, there are several issues with the paper with regard to severe safety violations during training, compounding errors due to model error and policy error, and lack of experiments + comparisons. As such, I don't think the paper is ready for publication, and does not provide concrete reliable takeaways.

---

> ### Author Response · Authors · 2021-11-17
> **Author Response (Part 1 of 3)**
>
> Thank you for your very detailed review. We are glad you found our work intuitive. We address the concerns raised below.
>
> `An important limitation of the approach is that since visiting unsafe states are discouraged during exploration (by penalizing the agent with -negative rewards), the agent is unlikely to visit unsafe states, which is good for being safety but bad for learning an accurate dynamics model in those regions. Hence, dynamics model rollouts are likely to be very inaccurate around unsafe parts of the state-space, thereby making the planning process inaccurate. `
>
> We agree that unsafe states being discouraged during exploration can lead to an inaccurate world model being learnt; however, this needs to be traded-off with the need of being somewhat safe during the exploration period. In the paper, we address this problem with the novel notion of shield introduction schedules (to recap, this basically means turning shielding off at certain times to allow for world model learning), as well as a simple heuristic for calibrating such schedules based specifically on how well violation states are modelled (see Appendix G). In effect, this gives the agent time to learn about violation states before punishing it for visiting them and, empirically, leads to no degradation in the quality of the latent model around unsafe parts of the state space (see Appendix G for a qualitative comparison between the unshielded and shielded agents' latent dynamics models).
>
> We further argue that the modelling of dynamics within unsafe regions is not too important as the goal of the agent is to avoid such regions. Therefore, it is perhaps more important to learn how to avoid the transition into unsafe regions than it is to learn the dynamics within unsafe regions.
>
> `Another limitation of the approach is that it necessarily requires visiting a lot of unsafe states during training, for learning the world model, and hence can never be completely safe. This is also reflected in the results in Table 1 where the training violations are significantly high. It would be helpful to have a discussion about acknowledgement of this limitation in the paper. `
>
> You are absolutely right – visiting unsafe states is a necessary sacrifice when learning a notion of safety from scratch though, as you are likely aware, this is a sacrifice made by many methods in the safety-aware RL literature (e.g. Achiam et al., 2017; Srinivasan et al., 2020; Yang et al., 2020). We agree that this should be mentioned in the paper and have updated the discussion section of the manuscript accordingly.
>
> ` There are no guarantees of how accurate the safety shield is given errors in the learned dynamics model. It will be helpful to formally quantify the safety constraint satisfaction as some function of model-error, or provide intuitions about the extent to which inaccuracies in the model are likely to affect safety violations. `
>
> We agree that it would be helpful to have formal bounds on performance as some function of errors in the learned dynamics model. However, we believe that such a problem is non-trivial and would deserve a paper or two in its own right (especially given our page limit of 9 pages). For example, we would need to be able to quantify model error in a more meaningful way than standard error metrics and consider questions such as “what’s more deleterious to an agent’s performance: wild inaccuracies in modelling a region of the state space that the agent does not visit or small inaccuracies in a region of the state space that the agent regularly traverses?”.
>
> Notwithstanding, the main goal of this paper is to introduce the foundations of latent shielding so that the community at least has a basis to work on interesting problems such as the one proposed in your above comment.
>
> `Since in equation 4 outputs a "safe" action by rejection sampling, there is no guarantee that a safe action if it exists will be found. This is because the learned model is inaccurate, so at any given time, when a safe action is not found, we can't be sure whether this is because a safety violation is inevitable, or because there exists a safe action which is not feasible under the current (inaccurate) dynamics. `
>
> This is a valid concern. In our work, we address this through shield introduction schedules. Discussed in more detail above, these schedules allow an accurate model to be learned before the shield is allowed to interfere. Furthermore, as mentioned in the Discussion, one advantage of our method over model-free safety-aware RL is that it is possible to visualise the trajectories predicted by the learned latent dynamics model, thus providing some explainability. These visualisations can, in turn, be used in development to help debug the world model; or in a production environment as part of a human-in-the-loop setup. Please see Appendix G for an example of such visualisations.

---

> > ### Author Response · Authors · 2021-11-17
> > **Author Response (2 of 3)**
> >
> > `There are no comparisons to model-free safe RL approaches (for example the constrained optimization and safety critics papers cited in the 2nd paragraph of the introduction). In comparison to prior safe RL works that learn a safe policy without learning a safe dynamics+rewards model i.e. in comparison to safe model-free approaches, the proposed approach will likely suffer from a compounding of errors. This is likely to happen because the proposed method has a large number of learned components (parts of the model) and relies on the model being accurate for planning. Hence, a comparison with some prior model-free safe RL methods is important. `
> >
> > We have updated the manuscript with some experiments using CPO (Achiam et al., 2017) as a baseline. A disadvantage of model-free RL is that they are often far less sample-efficient than model-based approaches (which is a point that Hafner et al., 2019a,b emphasise very much). This disadvantage manifests itself clearly in the experiments – for instance, even after training for 5x more steps on the low-dimensional CD environment, CPO fails to even meet the performance of the unshielded Dreamer agent, despite having incurred many more violations during training.
> >
> > Regarding the compounding of errors, Hafner et al., 2019b (which introduces RSSMs) propose a method for ensuring the accuracy of multi-step predictions thus reducing the likelihood of catastrophic compounding of errors (see “Latent Overshooting”). Empirical evaluation by Hafner et al., 2019b as well as later work in Hafner et al., 2019a empirically demonstrate robustness to the compounding of errors. Therefore, at least from an empirical standpoint, this seems to be a reasonable trade-off to make.
> >
> > `A very important limitation is the lack of sufficient experiments. Experiments are only on toy settings. It is necessary to evaluate safe behavior on more complex dynamical systems (for example robot manipulation, locomotion, -environments). Many of the prior safe RL papers cited evaluate on such environments. `
> >
> > Having considered all reviewers’ comments on the experimental evaluation as well as our computational limitations, we have decided to add a model-free baseline from Achiam et al., 2017. This does, unfortunately, mean that we could not add another more complex environment. We do believe, however, that the environments used are sufficient to demonstrate the two key points that we set out to show: that latent shielding can deal with high-dimensional visual environments (which many safety-aware RL approaches already struggle with), as well as stochasticity in the environmental dynamics.
> >
> > `[clarification] Is the unshielded variant (in blue in the plots), normal Dreamer? If so, then how does the proposed method compare with Dreamer in the normal DeepMind Control Suite environments? It is necessary to see this in order to understand whether the proposed method achieves safe behavior while compromising on task performance in environments where safety is not explicitly important. `
> >
> > Yes, the unshielded variant is just a vanilla Dreamer agent.
> >
> > In the normal DMC suite environments without an explicit safety signal, we see no reason for our agent to perform differently to an unshielded agent. This is because the shield is essentially a wrapper around the agent that interferes when an unsafe action is suggested by the agent’s unshielded policy. If no explicit safety signal is given, then the shield will quickly learn never to interfere, and the agent’s behavior boils down to that of an unshielded agent.
> >
> > Indeed, that is not to say that our method can’t be adapted to incorporate certain priors into the RL training process (such as in Icarte et al., 2018) and expedite learning (e.g. to signal to a quadrupedal robot that being upside-down is not a good idea). Nevertheless, we believe that such experiments would stray too far from the main focus of this work: safety.
> >
> > `[minor] I am curious if the title has any relevance to the contents of the paper? I understand that it is a spin-off of "Do Androids Dream of Electric Sheep" and the connection to the paper in my understanding is: Dream --> Dreamer; and Fences --> Shield. If this is the case / there is more to it, then a one line explanation of it somewhere in the paper / in the appendix might be helpful. `
> >
> > Done! See Appendix J.

---

> > > ### Author Response · Authors · 2021-11-17
> > > **Author Response (Part 3 of 3)**
> > >
> > > We hope that this has adequately addressed your concerns. If not, we'd love to hear about any remaining concerns in order to improve the paper as much as possible.
> > >
> > > **References**
> > >
> > > Joshua Achiam et al. Constrained Policy Optimization. In Proceedings of the 34th International Conference on Machine Learning, volume 70 of Proceedings of Machine Learning Research, pp. 22–31. PMLR, 2017.
> > >
> > > Danijar Hafner et al. Dream to Control: Learning Behaviors by Latent Imagination. In International Conference on Learning Representations, 2019a.
> > >
> > > Danijar Hafner et al. Learning Latent Dynamics for Planning from Pixels. In Proceedings of the 36th International Conference on Machine Learning, pp. 2555–2565. PMLR, 2019b.
> > >
> > > Rodrigo Toro Icarte et al. Teaching Multiple Tasks to an RL Agent using LTL. In Proceedings of the 17th International Conference on Autonomous Agents and MultiAgent Systems, pp. 452-261. 2018.
> > >
> > > K. Srinivasan, et al. Learning to be Safe: Deep RL with a Safety Critic. ArXiv, abs/2010.14603, 2020.
> > >
> > > Tsung-Yen Yang et al. Projection-based Constrained Policy Optimization. In International Conference on Learning Representations, 2020.

---

> > > > ### Comment · Reviewer_eXDd · 2021-11-21
> > > > **Thanks for the detailed response. My concerns unfortunately still remain.**
> > > >
> > > > I thank the authors for their detailed response, and for updating the paper with comparison against CPO. Unfortunately my concerns regarding the safety and model accuracy aspects of the method still remain, and I would not lean towards acceptance for the current paper.
> > > >
> > > > Since the proposed approach is a minor modification of Dreamer to account for safety, the contribution for safety should be more thoroughly analyzed in my opinion. Yes, prior safe RL approaches are also not 100% safe during training, but most of them are model-free methods, and do not attempt to learn an accurate dynamics model (+ a policy based on it). I am still unclear about the tradeoffs between safe exploration enforced by the latent shielding, and the tradeoff for model-accuracy around (not inside) unsafe regions. Latent shielding as originally proposed in Dreamer and Planet doesn't account for constrained exploration by the learned policy - so I don't think the benefits of that directly apply here. The authors mentioned that doing this would be a lot of work during the rebuttal period, which I agree with - but it will significantly help improve the paper and make the significance of its contributions clear. Related to this are the evaluations on more complex dynamics systems, as also pointed out by the other reviewers, which are important to understand the benefits of the approach in more realistic settings where talking about safety makes sense. Kindly take into account this feedback and those from the other reviewers for a major revision of the paper.

---

### Official Review · Reviewer_HtqJ · 2021-11-02

**Correctness:** 4
**Technical Novelty And Significance:** 3
**Empirical Novelty And Significance:** 3
**Recommendation:** 8
**Confidence:** 3

**Main Review:**

The paper is well-written and structured and accessible to the reader.

The new latent shielding has one major drawback and weakness, which is of course mention and acknowledged in the paper, but still a concern to some degree: The guarantees that other shielding techniques have are lost as the method only works on the latent model.
This is advantage and disadvantage at the same time. The advantage is the non-requirement of hand-crafted rules. However, violation information is required during training which also needs to be modeled from somewhere, so the experts are still needed, although it might be easier to identify a violation than to mitigate it from rules. The disadvantage is that the model cannot be deployed alone for RL safety, but at least for other applications or as the main safety technique with an additional external shield as fallback.

I'm a bit disappointed by the alternative action in case of a violation. Here I would expect that the world model would allow to perform better planning than just selecting any non-violating action.

The experimental evaluation is okay. A more complicated environment could have been chosen to really highlight the advantage of not needing hand-crafted rules could have been chosen, e.g. car racing or also one from Alshiekh 2018 for comparison.

The method itself is sound, although it is based on many existing contributions and it is a judgement call whether the contributions over the existing techniques are sufficiently high, but I think they are and it is an interesting contribution and basis for future work.

**Summary Of The Paper:**

The paper introduces shielding technique for RL safety into world model RL agents, like the Dreamer model.
The shielding technique works on the latent space and is trained from violation information only, i.e. without the hand-crafted design of safety rules.
An experimental evaluation is performed on two environments and compared against the unshielded version and BPS from the literature.

**Summary Of The Review:**

The paper explores an interesting direction for RL safety and shielding and even though it cannot provide performance guarantees, it is helpful to the agent's performance and reduces some violations.
The contribution will be helpful to the community.

---

> ### Author Response · Authors · 2021-11-17
> **Author Response (Part 1 of 1)**
>
> Thank you for your detailed review. We are glad you have had a positive impression of the work. We address a couple of the points raised below.
>
> `Violation information is required during training which also needs to be modeled from somewhere, so the experts are still needed, although it might be easier to identify a violation than to mitigate it from rules. `
>
> The specification used by the shield will indeed need to be written by human experts; however, we believe that (from experience) it is generally easier than formally specifying the environmental dynamics. Moreover, the formal verification community has a rich literature on efficiently monitoring systems at runtime with respect to safety specifications.
>
> `The disadvantage is that the model cannot be deployed alone for RL safety, but at least for other applications or as the main safety technique with an additional external shield as fallback. `
>
> We agree with this assessment, though would like to emphasise that our approach is not the only one that faces such a limitation. Many approaches (e.g. Achiam et al., 2017; Yang et al., 2020), especially outside of the formal verification and shielding literature, still see safety violations when evaluated on relatively low-dimensional environments – they instead aim to reduce the number of violations as opposed to eliminating them absolutely. In order to obtain formal safety guarantees, it would very likely be necessary that the learned environmental dynamics model be formally verified. Such a task, however, is non-trivial and thus, given how much we already introduce in this work, left as a problem for future research. The main goal of this paper is to introduce the foundations of latent shielding so that the community at least has a basis to work from.
>
> `I'm a bit disappointed by the alternative action in case of a violation. Here I would expect that the world model would allow to perform better planning than just selecting any non-violating action. `
>
> We chose to use simple rejection sampling as our alternative policy as we thought it was the best way to demonstrate the core approach. Of course, other policies are possible (such as choosing the “next most-preferred action”) have been tried and extensively evaluated in other works (Alsheikh et al., 2018; Srinivasan et al., 2020). Notwithstanding, even with such a simple alternative policy, our approach is competitive with the existing literature.
>
> `A more complicated environment could have been chosen to really highlight the advantage of not needing hand-crafted rules could have been chosen, e.g. car racing or also one from Alshiekh 2018 for comparison. `
>
> Having considered all reviewers’ comments on the experimental evaluation as well as our computational limitations, we have decided to add a model-free baseline from Achiam et al., 2017. This does, unfortunately, mean that we could not add another environment. We do believe, however, that the environments used are sufficient to demonstrate that latent shielding can deal with high-dimensional visual environments (note that in Alsheikh et al., 2018, the visual environment’s shield state is derived from privileged access to the emulator whereas in the VGW environment, the shield’s state is derived from observation), as well as some stochasticity in the environmental dynamics (with sticky actions in the CD environment).
>
>
> We hope that this has adequately addressed your concerns. If not, we'd love to hear about any remaining concerns in order to improve the paper as much as possible.
>
>
> **References**
>
> Joshua Achiam, David Held, Aviv Tamar, and Pieter Abbeel. Constrained Policy Optimization. In Proceedings of the 34th International Conference on Machine Learning, volume 70 of Proceedings of Machine Learning Research, pp. 22–31. PMLR, 2017.
>
> Mohammed Alshiekh et al. Safe Reinforcement Learning via Shielding. In Proceedings of the AAAI Conference on Artificial Intelligence, volume 32, 2018.
>
> K. Srinivasan, et al. Learning to be Safe: Deep RL with a Safety Critic. ArXiv, abs/2010.14603, 2020.
>
> Tsung-Yen Yang et al. Projection-based constrained policy optimization. In International Conference on Learning Representations, 2020.

---

> > ### Comment · Reviewer_HtqJ · 2021-11-25
> > **Response**
> >
> > Dear authors,
> >
> > thank you for your response and the revision of the manuscript.
> > I acknowledge that it addresses some of my concerns and questions and is an improvement over the initial submission.
> >
> > I understand that you cannot solve everything in the problem domain in one paper and that similar restrictions apply in other papers. However, I also see that these restrictions need to either be explicitly addressed or overcome to actually advance the field.
> >
> > Nevertheless, to acknowledge the improvements, I'm going to increase my score.

---

### Official Review · Reviewer_WV6o · 2021-11-02

**Correctness:** 3
**Technical Novelty And Significance:** 2
**Empirical Novelty And Significance:** 2
**Recommendation:** 5
**Confidence:** 3

**Main Review:**

Overall, this was an interesting paper, but I have several concerns regarding the experiments.
 - One issue is that the difference in performance between “Latent” and “Unshielded” does not seem significant (i.e., the error bars are substantially overlapping in Figure 3 and in several rows of Table 1).
 - It’s difficult to interpret the numerical values of the test rewards. It would be helpful to report more interpretable performance metrics, such as the fraction of targets reached in the VGW environment, and the time-to-edge-of-cliff in the CD environment.
 - “Unshielded” does not seem like a fair baseline, since it does not make use of the labelling function (while the “Latent” method has access to the labelling function at training time). A stronger baseline might be [Trial Without Error](https://arxiv.org/abs/1707.05173), which also trains a classifier to predict whether or not states are unsafe using labels from an oracle (in this case, a human supervisor), and trains an RL agent to avoid unsafe states.

I’m also concerned about the novelty of the method: it is essentially Dreamer, but with a safety constraint on the planned trajectories. I may have missed the algorithmic contribution of this paper, and am happy to revise this comment during the discussion period.

Update
----
Thank you to the authors for the thoughtful response. I have increased my score.

**Summary Of The Paper:**

The paper proposes a safe model-based reinforcement learning algorithm that trains a classifier to predict whether or not a state is unsafe, and plans trajectories that avoid unsafe states. The key idea is to train the classifier on latent representations of states given by the latent dynamics model in a recurrent state-space model. Experiments in simulated navigation tasks show that in some environments, the proposed method achieves higher reward and violates safety constraints less often than an ablated variant that does not constrain planned trajectories to be safe.

**Summary Of The Review:**

Weak experimental results and lack of comparisons to prior methods

---

> ### Author Response · Authors · 2021-11-17
> **Author Response (Part 1 of 2)**
>
> Thank you for your detailed and thoughtful review. We address your concerns below:
>
> `One issue is that the difference in performance between “Latent” and “Unshielded” does not seem significant (i.e., the error bars are substantially overlapping in Figure 3 and in several rows of Table 1). `
>
> We acknowledge that there is some overlap in the error bars for the “Latent” (our agent) and “Unshielded” reward curves, especially towards the end of training (Figure 3). However, we do not believe this to be a drawback as the goal of the paper is to achieve reasonable performance while minimising safety violations. The observation that there is little difference in reward thus suggests that the task performance of the agent (as measured by reward) does not take too big of a hit despite the agent behaving in a safer way.
>
> This is also true for Table 1 in which our agent gives similar test-time rewards to its unshielded counterpart while consistently also achieving a more-than-sevenfold reduction in violations on the VGW environment and a threefold reduction on the stochastic CD environment.
>
> To summarize, while a direct comparison of reward collected by the agents may lead one to conclude that the experimental results are weak, we believe that looking at the results through the context of the paper (i.e. that we achieve a sizeable decrease in violations while making sure that the difference in reward “does not seem significant”) leads to a much more compelling interpretation (which is indeed the one that we’re going for).
>
> `It’s difficult to interpret the numerical values of the test rewards. It would be helpful to report more interpretable performance metrics, such as the fraction of targets reached in the VGW environment, and the time-to-edge-of-cliff in the CD environment. `
>
> Thank you for pointing this out! We have updated the manuscript with the suggested metrics – please see Table 2 in Appendix G.

---

> > ### Author Response · Authors · 2021-11-17
> > **Author Response (Part 2 of 2)**
> >
> >
> >  `“Unshielded” does not seem like a fair baseline, since it does not make use of the labelling function (while the “Latent” method has access to the labelling function at training time). A stronger baseline might be Trial Without Error, which also trains a classifier to predict whether or not states are unsafe using labels from an oracle (in this case, a human supervisor), and trains an RL agent to avoid unsafe states. `
> >
> > We agree that benchmarking our safety-aware agent against an agent designed without safety as a priority is unfair. This is why we also benchmark our agent on a method from the formally verified safe RL literature, bounded prescience shielding (BPS) (Giacobbe et al., 2021). In the deterministic case, BPS can be thought of as having the same effect as a “Blocker” from the Trial Without Error paper, where the classifier error is zero (since the BPS is provably H-bounded safe, unlike the CNN used to implement the “Blocker”). We thus believe that BPS is in fact a stronger baseline than Trial Without Error.
> >
> > We would also like to take the opportunity to quickly highlight a major advantage of our method over Trial Without Error: that we do not need human supervision. This leads to time savings (the authors mention it takes 4.5h of human supervision to train an agent to safely play Pong), as well as eliminating the chance that the agents “learn unsafe habits” from their human supervisors.
> >
> > Finally, in light of other reviewers’ suggestions to add a model-free baseline, we have added experiments with CPO (Achiam et al., 2017). We invite you to take a look at our updated Table 1.
> >
> >    `I’m also concerned about the novelty of the method: it is essentially Dreamer, but with a safety constraint on the planned trajectories. I may have missed the algorithmic contribution of this paper, and am happy to revise this comment during the discussion period. `
> >
> > Thank you for being upfront with this concern. We’ll try to address it from two angles: conceptual and technical.
> >
> > Our main conceptual contribution is that of latent shielding. Latent shielding is conceptually novel in that it combines the deep latent world models and shielding literature to address the problem of requiring a handcrafted abstraction of the environment, a pertinent problem for the shielding community as mentioned by Anderson et al., 2020, Giacobbe et al., 2021 as well as others. To our knowledge, few, if any, works have attempted to tackle the problem from this angle.
> >
> > Our work’s technical contribution is mostly focused around a particular implementation of latent shielding based on Dreamer and can be split into three main points. The first, as you rightly point out, is the modification of Dreamer’s RSSM to incorporate safety-awareness. In addition to this, however, we also propose an adaptation of bounded prescience shielding, ABPS. Notwithstanding the fact that ABPS operates within our learned latent abstraction (which, to our knowledge, is among the shielding literature’s only forays into not using a hand-crafted abstraction), it introduces a new sampling procedure for the system proposed in Giacobbe et al., 2021 that reduces the number of samples required at each timestep. This can be seen most dramatically in the experiments in the CD environments where 20 samples were sufficient to be on par with bounded prescience shielding, which required 1024 samples. Thirdly, motivated by the observation that too much shielding early on in training can lead to a poor world-model, we introduce a schedule for shielding during training. To our knowledge, the observation that shielding can be deleterious to the training of deep model-based agents has not yet appeared in the literature, let alone a method of preventing it (which we propose).
> >
> > All in all, our work draws on an exciting range of ideas from existing works and proposes a foundational solution to a pertinent problem in the shielding literature. We thus believe it to be a reasonable contribution to the shielding community that others will use as a springboard to explore shielding with learned abstractions.
> >
> >
> > We hope that this has adequately addressed your concerns. If not, we'd love to hear about any remaining concerns in order to improve the paper as much as possible.
> >
> > **References**
> >
> > Mirco Giacobbe et al. Shielding Atari Games with Bounded Prescience. In Proceedings of the 20th International Conference on Autonomous Agents and MultiAgent Systems, pp. 1507–1509, 2021.
> >
> > Joshua Achiam, David Held, Aviv Tamar, and Pieter Abbeel. Constrained Policy Optimization. In Proceedings of the 34th International Conference on Machine Learning, volume 70 of Proceedings of Machine Learning Research, pp. 22–31. PMLR, 2017.
> >
> > Greg Anderson, Abhinav Verma, Isil Dillig, and Swarat Chaudhuri. Neurosymbolic Reinforcement Learning with Formally Verified Exploration. In Advances in Neural Information Processing Systems, volume 33, pp. 6172–6183, 2020.

---

### Official Review · Reviewer_Lq99 · 2021-11-04

**Correctness:** 2
**Technical Novelty And Significance:** 2
**Empirical Novelty And Significance:** 2
**Recommendation:** 5
**Confidence:** 4

**Main Review:**

The main contribution is minor. The proposed approach is mostly is a straightforward combination of existing methods. The experiments are limited in complexity and thoroughness. The baseline methods such as Dreamer (Hafner 2021) have been evaluated over a wide range of test scenarios. Therefore two environments for evaluation are not sufficient. Furthermore, for fair comparison methods from Safe Reinforcement Learning should also be included.

**Summary Of The Paper:**

This paper proposes a safe Reinforcement Learning approach using Latent Shielding. The agent learns a  data-driven recurrent state-space model representing a world model based on an existing approach (Hafner et al., 2019b). The learned latent world model allows agents to foresee various trajectories arising from different actions and avoid unsafe states. In their setting, the environment also provides binary labeling over states indicating an occurrence of violation in each step, formulated as syntactically co-safe Linear Temporal Logic (scLTL) (Kupferman & Vardi, 2001). The proposed approach is evaluated on two environments (1) Visual Grid World and (2) Cliff driver and compared against Dreamer agent with no shielding (Hafner et al., 2019a; 2021) and a Dreamer agent with Bounded Prescience Shield (Giacobbe et al., 2021).


**Summary Of The Review:**

The paper contribution is minor. Their experiments are limited in terms of test environments and benchmark methods.

---

> ### Author Response · Authors · 2021-11-17
> **Author Response (Part 1 of 2)**
>
> Thank you for your review. We address the concerns raised below.
>
> `The proposed approach is mostly a straightforward combination of existing methods. `
>
> Our work takes ideas from both the shielding and world models literature so we appreciate that it may superficially seem that we are just picking ideas from their respective fields and crudely gluing them together. This, however, is far from the case.
>
> The centerpiece of our work is the concept of latent shielding, a novel framework for addressing the problem of requiring a handcrafted abstraction of the environment. This is a pertinent problem for the shielding community as mentioned by a number of works including Anderson et al., 2020 and Giacobbe et al., 2021. To our knowledge, few, if any, works have attempted to tackle the problem from this angle.
>
> Our work’s technical contribution is mostly focused around a particular implementation of latent shielding based on Dreamer. Like all shields, our approach involves wrapping the agent in a reactive system (the shield) that blocks unsafe actions. However, unlike previous approaches, this wrapper is based on a data-driven abstraction of the environment: in particular, an augmented version of the RSSM proposed by Hafner et al., 2019b that encodes safety into state representations.
>
> Moreover, the manner in which the shield queries its abstraction is novel, taking a heuristic sampling-based approach as opposed to exhaustive search. The effects of this change can be seen most dramatically in the experiments in the CD environments where 20 samples were sufficient to be on par with bounded prescience shielding, which required 1024 samples.
>
> Finally, even with these components in place, training the new shielded agents can prove non-trivial, as pointed out by Reviewer #4 (eXDd), with potential problems arising with exploration and the trade-off between safety and learning accurate dynamics. Motivated by these problems and the observation that too much shielding early on in training can be detrimental to the learning process, we introduce the new notion of schedules for shielding during training. To our knowledge, the observation that shielding can be deleterious to the training of deep model-based agents has not yet appeared in the literature, let alone a method of preventing it (which we propose).
>
> To summarise, our work draws on an exciting range of recent ideas from existing works and proposes a foundational solution to a pertinent problem in the shielding literature. We thus believe it to be a reasonable contribution to the shielding community that others will use as a springboard to explore shielding with learned abstractions.
>
>
>
> `The experiments are limited in complexity and thoroughness. The baseline methods such as Dreamer (Hafner 2021) have been evaluated over a wide range of test scenarios. Therefore two environments for evaluation are not sufficient.  `
>
> We acknowledge that the number of environments we evaluate on is few compared that of previous works such as Dreamer (Hafner et al., 2019a). This is due to computational limitations on our side – reproducing even a single training run through the continuous control environments from Hafner et al., 2019a on our single-GPU rig would take over a week, based on the training details provided in the paper (3h per 1e6 training steps * 1e8 steps = 300h).
>
> We do believe, however, that the environments used are sufficient to demonstrate the two key points that we set out to show: that latent shielding can deal with high-dimensional visual environments (which many safety-aware RL approaches already struggle with), as well as stochasticity in the environmental dynamics. Furthermore, we argue these experiments are in fact conducted in a thorough manner: training for each agent is repeated over 5 different random seeds, which is in line with the evaluation procedure in Hafner et al., 2019a. We also provide training reward and violation plots with error bars (Figure 3 and Appendix E), test-time metrics with standard deviations each evaluated over 50 runs (Table 1), as well as visualisations of the agents’ latent dynamics models (Appendix F).

---

> > ### Author Response · Authors · 2021-11-17
> > **Author Response (Part 2 of 2)**
> >
> >
> > `For fair comparison methods from Safe Reinforcement Learning should also be included. `
> >
> > As mentioned in the summary of the paper, we evaluate our agent against a baseline of both a vanilla Dreamer agent (Hafner et al., 2019a) and a Dreamer agent with bounded prescience shielding (Giacobbe et al., 2021), the latter of which is a recent method from the safe RL literature. We chose to use bounded prescience shielding as a baseline as it is the method conceptually closest to our work, thus making it highly appropriate. Nevertheless, taking into account suggestions from the other reviewers to add a model-free baseline, we have updated the manuscript to include experiments with CPO (Achiam et al., 2017), a well-known method from the model-free safe RL literature which we outperform. We invite you to take a look at our updated manuscript – in particular, Table 1.
> >
> > We hope that this has adequately addressed your concerns. If not, we'd love to hear about any remaining concerns in order to improve the paper as much as possible.
> >
> > **References**
> >
> > Mirco Giacobbe et al. Shielding Atari Games with Bounded Prescience. In Proceedings of the 20th International Conference on Autonomous Agents and MultiAgent Systems, pp. 1507–1509, 2021.
> >
> > Greg Anderson et al. Neurosymbolic Reinforcement Learning with Formally Verified Exploration. In Advances in Neural Information Processing Systems, volume 33, pp. 6172–6183, 2020.
> >
> > Joshua Achiam et al. Constrained Policy Optimization. In Proceedings of the 34th International Conference on Machine Learning, volume 70 of Proceedings of Machine Learning Research, pp. 22–31. PMLR, 2017.
> >
> > Danijar Hafner et al. Dream to Control: Learning Behaviors by Latent Imagination. In International Conference on Learning Representations, 2019a.
> >
> > Danijar Hafner et al. Learning Latent Dynamics for Planning from Pixels. In Proceedings of the 36th International Conference on Machine Learning, pp. 2555–2565. PMLR, 2019b.

---

> > > ### Comment · Reviewer_Lq99 · 2021-11-19
> > > **Review updated**
> > >
> > > I have reviewed the modifications. The addition of CPO is appreciated, and considering authors have limited access to computation resources. I have revised my score. However, I believe the paper is still below the acceptance bar. The framework has a minor novelty and is mainly based on existing approaches. The evaluation section could also benefit from adding more complicated environments.

---

### Decision · Program_Chairs · 2022-01-20

**Decision:**

Reject

**Comment:**

The paper proposes a modification of the Dreamer method to account for safety constraints.
There is agreement among the majority of reviewers that the paper lacks in novelty (with respect to Dreamer), in the safety analysis, and in convincing experiments.
I encourage the authors to take the detailed reviews into account when improving their work.